# Distinct Gut Microbial Enterotypes and Functional Dynamics in Wild Striped Field Mice (*Apodemus agrarius*) across Diverse Populations

**DOI:** 10.3390/microorganisms12040671

**Published:** 2024-03-28

**Authors:** Yongzhen Wu, Taoxiu Zhou, Shengmei Yang, Baofa Yin, Ruiyong Wu, Wanhong Wei

**Affiliations:** College of Bioscience and Biotechnology, Yangzhou University, Yangzhou 225009, China; mx120180800@yzu.edu.cn (Y.W.); zhoutaoxiu@163.com (T.Z.); smyang@yzu.edu.cn (S.Y.); bfyin@yzu.edu.cn (B.Y.); wuruiyong@yzu.edu.cn (R.W.)

**Keywords:** *Apodemus agrarius*, gut microbiota, enterotypes, geographic populations, community assembly

## Abstract

Rodents, including the striped field mouse (*Apodemus agrarius*), play vital roles in ecosystem functioning, with their gut microbiota contributing significantly to various ecological processes. Here, we investigated the structure and function of 94 wild *A. agrarius* individuals from 7 geographic populations (45°57′ N, 126°48′ E; 45°87′ N, 126°37′ E; 45°50′ N, 125°31′ E; 45°59′ N, 124°37′ E; 46°01′ N, 124°88′ E; 46°01′ N, 124°88′ E; 46°01′ N, 124°88′ E), revealing two distinct enterotypes (Type1 and Type2) for the first time. Each enterotype showed unique microbial diversity, functions, and assembly processes. Firmicutes and Bacteroidetes dominated, with a significant presence of *Lactobacillus* and *Muribaculaceae.* Functional analysis highlighted metabolic differences, with Type1 emphasizing nutrient processing and Type2 showing higher energy production capacity. The analysis of the neutral model and the null model revealed a mix of stochastic (drift and homogenizing dispersal) and deterministic processes (homogenous selection) that shape the assembly of the microbiota, with subtle differences in the assembly processes between the two enterotypes. Correlation analysis showed that elevation and BMI were associated with the phylogenetic turnover of microbial communities, suggesting that variations in these factors may influence the composition and diversity of the gut microbiota in *A. agrarius*. Our study sheds light on gut microbial dynamics in wild *A. agrarius* populations, highlighting the importance of considering ecological and physiological factors in understanding host–microbiota interactions.

## 1. Introduction

The concept of “enterotype” has emerged as a key framework for understanding the complex interplay between gut microbial composition, host factors, and environmental influences and was first conceptualized in extensive studies of the human gut microbiome [1,2]. It represents distinct patterns of microbial communities within the intestinal tract, each associated with host characteristics and potentially related to various phenotypes, including diseases [3]. For example, people with an enterotype marked by *Bacteroides*, which efficiently breaks down carbohydrates to help the host obtain energy from food, tend to eat a high-fat, high-protein diet [3,4]. People with marked enterotypes, which tend to be associated with non-Western diets, tend to have plant-based carbohydrates [3,5]. And people with the *Ruminococcus* marked enterotype tend to be high in carbohydrates; In addition, *Ruminococcus* helps with sugar absorption; so, people with this enterotype may suffer more weight problems [3,6].

Despite initial beliefs in the stability of enterotypes over time, subsequent research has highlighted their susceptibility to environmental factors. Diet plays an essential role in shaping enterotype patterns, with long-term dietary habits being intricately linked to specific gut microbial profiles [7,8], and short-term dietary adjustments can lead to rapid changes in gut microbiota composition [9]. Furthermore, the use of antibiotics can cause temporary or permanent changes in the gut microbiota, while the consumption of probiotics can make the enterotype more stable [10]. Age is another multifaceted factor influencing enterotype patterns, necessitating separate considerations during analysis [11,12].

Clustering analyses based on indices such as the Calinski–Harabasz index have been instrumental in identifying enterotypes, which were processed by folding highly multidimensional niche microbiomes into a small number of categories [3]. The identification of enterotypes has not only facilitated the analysis of host–microbe interactions, but has also been extended to studies involving other animals, such as chimpanzees (*Pantroglodytes schweinfurthii*) [13], pigs (*Sus scrofa domesticus*) [14,15], buffaloes (*Syncerus caffer*) [16], mice (*Mus musculus*) [17], bumblebees (*Bombus* sp.) [18], fruit flies (*Drosophila* sp.) [19], and pikas (*Ochotona curzoniae*) [20,21,22], suggesting that enterotypes may be prevalent in animal gut microbes and may help to better characterize these microbiomes and to provide answers to questions about the correlation between microbes and the host.

Ecological theories, including niche differentiation and neutral processes, offer valuable insights into the determinants and dynamics of enterotypes [23]. In microbial communities, niche differentiation refers to the process whereby distinct microbial populations adapt and specialize within a shared environment, exploiting different ecological niches characterized by varied resource availability and environmental conditions. This entails the evolution of diverse metabolic pathways, growth characteristics, and survival strategies among microbial populations to optimize resource utilization and environmental fitness. Such differentiation serves to mitigate interspecies competition, thereby fostering microbial community diversity and stability. Niche theory assumes that deterministic factors such as species characteristics, interspecies interactions, and environmental conditions control community structure and metabolic function [24]. That is, microbial communities are formed by deterministic biological factors (species interactions, such as competition and predation) and abiotic factors (environmental factors, such as pH and temperature), which are caused by different habitat preferences and adaptations of microorganisms [25].

Neutral processes, on the other hand, refer to ecological dynamics that are governed primarily by random or stochastic factors rather than deterministic forces [25,26]. Unlike niche-based processes, which emphasize the importance of species traits and environmental conditions in shaping community structure, neutral processes operate under the assumption of ecological equivalence among species. The neutral process theory assumes that microbial loss and gain show a random balance in taxa, that is, random processes (birth, death, migration, immigration, speciation, and limited dispersal) shape the structure of the microbial community [27,28].

Community assembly is a necessary process for enterotype formation, which is structured and organized over time and encompasses various ecological mechanisms, including species interactions, environmental filtering, and colonization dynamics [29]. Community assembly processes are variable and different in different ecosystems [30]. For example, deterministic processes (such as variable selection) play a major role in driving the assembly of the stonefish microbiome, while random processes (such as drift) also play a role [31]. The aggregation of gut microbiota in pika is mainly determined by deterministic processes but has different contribution rates in different enterotypes [22]. In the Sable Island horse population, changes in the host microbiome are driven more by bacterial diffusion and ecological drift than by differential selection pressures [32]. These results suggest the importance of considering ecological processes in microbiome studies to understand aspects of the diversity, function, and biogeography of the microbial community.

The striped field mouse (*Apodemus agrarius*) is a globally distributed rodent species that has attracted significant attention due to its role as a carrier of up to 17 zoonotic pathogens [33,34]. Found in diverse geographical regions and ecosystems, the striped field mouse exhibits strong environmental adaptability and niche differentiation capabilities. Consequently, there is reason to believe that it may have different gut phenotypes, with the formation mechanisms of these phenotypes being closely linked to its ecological adaptability and behavioral patterns. This study aims to investigate the spatial distribution of gut microbiota in the striped field mouse and to explore the differences in ecological adaptability and host physiological characteristics among different gut phenotypes. By collecting gut samples from striped field mice in various geographical regions, we endeavor to identify distinct enterotypes and to analyze their relationships with host physiological traits, with the aim of providing new insights into the symbiotic relationship between gut phenotypes and hosts. We seek to address the following questions: (Ⅰ) How many types of gut microbiota communities can be identified in the striped field mouse? (Ⅱ) Are there differences in the composition and structure of microbial communities among different enterotypes? Do differences exist in microbial genomic functions? (Ⅲ) Are there differences in the assembly processes of microbial communities among different enterotypes? (Ⅳ) What impacts do factors such as altitude, body weight, body length, and body mass index have on the assembly of gut microbial communities in the striped field mouse?

## 2. Materials and Methods

### 2.1. Sample Collection

To increase sample size and representativeness, we combined 67 samples from 5 sampling sites (Dataset 1) and 27 samples from 2 sampling sites (Dataset 2) for overall analysis. The samples in Dataset 1 were collected from July to August 2020, and the locations of the five sites were: Dongsheng Village, Nangang District, Harbin City, Heilongjiang Province, China (S1: 45°57′ N, 126°48′ E, elevation 182 m, N = 10); Wanbao Town, Harbin City, Heilongjiang Province, China (S2: 45°87′ N, 126°37′ E, elevation 120 m, N = 24); Minzhu Village, Zhaoyuan County, Daqing City, Heilongjiang Province, China (S3: 45°50′ N, 125°31′ E, elevation 124 m, N = 15); Xinzhan Town, Zhaoyuan County, Daqing City, Heilongjiang Province, China (S4: 45°59′ N, 124°37′ E, elevation 139 m, N = 10); and Datong District, Daqing City, Heilongjiang Province, China (S5: 46°01′ N, 124°88′ E, elevation 132 m, N = 8). The samples in Dataset 2 were collected from July to August 2021 from two locations: Tumuji Town, Zhalaite Banner, Hinggan League, Inner Mongolia Autonomous Region, China (S6: 46°01′ N, 124°88′ E, elevation 160 m, N = 11), and Wuchagou Town, Arxan City, Hinggan League, Inner Mongolia Autonomous, China (S7: 46°01′ N, 124°88′ E, elevation 838 m, N = 16) (Appendix A; Figure 1). In both datasets, the collection and preservation methods for all the samples remained consistent. Detailed descriptions of the methods can be found in Wu et al. (2024) [35]. Specifically, in the northern agro-pastoral transitional zone of China, we established eight survey grids measuring 100 km × 100 km, each containing two sampling points. Among these points, we successfully captured individuals of *A. agrarius* at the seven locations previously mentioned. The distances between the survey sites exceeded 30 km. The samples were collected in July and August of 2020 (Dataset 1) and July and August of 2021 (Dataset 2).

At each survey site, we deployed small collapsible aluminum Sherman traps (2 × 2.5 × 6.5 inches) baited with peanut seeds, following a dusk-to-dawn trapping regime for two consecutive days. The captured animals were euthanized using ether and stored in sterile bags in a vehicle refrigerator at −20 °C. Cecal content samples were collected under non-laboratory conditions. To prevent contamination, all the dissecting tools were thoroughly cleaned with 75% ethanol, and between each sample collection, they were flame-sterilized using an alcohol lamp. Approximately 10 mg of cecal content was extracted from the distal end for microbial sampling. To preserve RNA integrity, the samples were placed in 4 mL of RNALater and stored in a vehicle refrigerator at −20 °C. Upon completion of the fieldwork, the samples were transported to the laboratory and stored at −80 °C before DNA extraction.

All the experimental procedures adhered to ethical guidelines, ensuring minimal impact on the subjects, and standardized laboratory protocols were followed to maintain data reliability and repeatability.

### 2.2. DNA Extraction and Sequencing

The extraction of DNA and amplification of 16S target fragments have been reported in previous studies. Briefly, total genome DNA from the samples was extracted using the CTAB method. The 16S rRNA genes of the V4 regions were amplified using the specific primers 515F (5′-barcode-GTGCCAGCMGCCGCGGTAA-3′) and 806R (5′-GGACTACHVGGGTWTCTAAT-3′). All the PCR reactions were carried out with 15 μL of Phusion^®^ High-Fidelity PCR Master Mix (New England Biolabs, Ipswich, MA, USA), 2 μM of forward and reverse primers, and about 10 ng of template DNA. The thermal cycling consisted of initial denaturation at 98 °C for 1 min, followed by 30 cycles of denaturation at 98 °C for 10 s, annealing at 50 °C for 30 s, and elongation at 72 °C for 30 s and finally at 72 °C for 5 min. Afterwards, we mixed the same volume of 1× loading buffer (containing SYBR green) with the PCR products and performed electrophoresis on 2% agarose gel for detection. The PCR products were mixed in equidensity ratios. Then, the mixture of PCR products was purified with the Qiagen Gel Extraction Kit (Qiagen, Hilden, Germany). Subsequently, sequencing libraries were generated using the TruSeq^®^ DNA PCR-free sample preparation kit (Illumina, San Diego, CA, USA) according to the manufacturer’s instructions, and index codes were added. Library quality was assessed using the Qubit@ 2.0 Fluorometer (Thermo Scientific, Waltham, MA, USA) and the Agilent Bioanalyzer 2100 system (Santa Clara, CA, USA). Finally, the libraries were sequenced on an Illumina NovaSeq platform to generate 250 bp paired-end reads.

### 2.3. Bioinformatics Analysis

The quality control of the demultiplexed paired-end sequence reads followed the protocol outlined in QIIME2 [36]. Initially, paired-end reads were assigned to respective samples on their unique barcodes; subsequently, they were truncated to remove barcodes and primer sequences. The merged reads were generated using FLASH (version 1.2.11, http://ccb.jhu.edu/software/FLASH/, accessed on 30 October 2021) [37], and the resulting spliced sequences were designated as raw tags. The quality filtering of these raw tags was carried out using Fastp software (version 0.20.0) to obtain clean, high-quality tags. These clean tags were then compared against the Silva database (https://www.arbsilva.de/, accessed on 30 October 2021) using Vsearch (version 2.15.0) to identify and remove chimeric sequences, yielding effective tags [38]. Following this, the effective tags underwent denoising using the DADA2 module within QIIME2 to generate initial amplicon sequence variants (ASVs). The ASVs with an abundance of fewer than 5 were filtered out [39]. To explore the phylogenetic relationships among the ASVs and the differences in dominant species between samples, multiple sequence alignments were performed using QIIME2. The absolute abundance of ASVs was normalized to a standard sequence number corresponding to the sample with the lowest number of sequences. Subsequent analyses of alpha and beta diversities were conducted on the normalized data output.

### 2.4. Random Forest Classifier Models

We used a random forest classifier (RFC) supervised learning algorithm in the R package “randomForest” [40] to screen for biomarkers that play an important role in enterotype classification. Using a ‘MeanDecreaseAccuracy’ metric, which quantifies the reduction in the prediction accuracy of random forest models when the values of a variable are randomly permuted, and a ‘MeanDecreaseGin’ metric, which measures the influence of each variable on the heterogeneity of the observation values at each node of a classification tree, thereby comparing the importance of the variables, to identify the important biomarkers. Subsequently, each model underwent cross-validation (10-fold), and receiver operating characteristic (ROC) curves were plotted.

### 2.5. Metagenome Prediction

The functional metagenomes of *A. agrarius* were predicted and analyzed according to the 16S rRNA gene through PICRUSt software (version 1.1.4) [41]. Then, the differences in gut gene functions (Wilcoxon rank sum test) in wild *A. agrarius* between the community clusters were calculated at level 1–3 using the existing group_significance.py command script through the QIIME platform.

### 2.6. Ecological Network Analysis

Network analyses were applied to reveal significant relationships between the relative abundance of ASVs. To reduce low-abundance or rare ASVs in our data, those ASVs with average relative abundance <0.01% were filtered. Then, the Spearman correlation values were computed among the ASVs. Robust correlations with the Spearman correlation coefficients >0.6 and false discovery rate corrected *p* < 0.05 were used to construct networks [42]. The nodes in the constructed network signify ASVs and the edges that link 2 ASVs mean the correlation values between 2 ASVs. The topological features of the ecological network were calculated using the “igraph” package. The subnetwork images were visualized with Graphviz software (version 2.38.0). To describe the topological properties, six topological features (i.e., network diameter, modularity, clustering coefficient, graph density, average degree, and average path length) at the network level were calculated using the ‘vegan’ and ‘igraph’ packages [43].

### 2.7. Bacterial Community Assembly Analyses

To infer the assembly processes of the microbial community of different enterotypes in *A. agrarius*, we evaluated the relative contributions of the stochastic and ecological processes using a neutral model and a null model. Firstly, following the approach of Burns et al. [26], we determined the distribution of ASVs within the 95% confidence interval predicted by the neutral model. The confidence interval was computed using the “hmisc” package in R (version 4.3.2). The goodness of fit of the neutral model was evaluated using the coefficient of determination (R^2^). Secondly, the ecological processes, including drift, selection, and dispersal, were assessed using null model analysis. We calculated the Raup–Crick index (RCI) and βNTI to quantify the relative contributions of these processes. A βNTI value ≥ 2 indicated the dominance of deterministic processes in the community assembly, while a value ≤ 2 suggested a predominant role of stochastic processes in shaping microbial communities. Subsequently, we combined βNTI and RCI to estimate the relative strengths of homogeneous selection (βNTI < −2), variable selection (βNTI > −2), homogeneous dispersal (RCI > 0.95 and βNTI < 2), dispersal limitation (RCI > 0.95 and βNTI < 2), and drift (|RCI| < 0.95 and |βNTI| < 2) in driving the microbial community. In addition, Mantel tests were employed to evaluate the relationship between the βNTI values and the environmental variables.

### 2.8. Statistical Analyses

All the statistical analyses and visualizations were conducted in R (version 4.3.2) unless otherwise specified. After subsampling to an even depth, the ASVs were merged by genus, resulting in 568 unique genera. To identify enterotypes in the microbiome data, we used the partition around medoids (PAM) algorithm based on Bray–Curtis distances in the R package ‘cluster’; then, the graphics such as PCoA were drawn. The optimal number of groups was chosen based on Calinski–Harabasz (CH) values [3]. Bray–Curtis distance calculations of genus abundance tables, principal coordinate analysis, and initial data visualization were performed by using the “phyloseq” package. We used ANOSIM similarity analysis to examine whether there were differences in microbial community structure among the different enterotypes of *A. agrarius*. The Wilcoxon rank sum test was used to compare the α-diversities, the relative abundance of microbial taxa, and the KEGG pathways between the different enterotypes. We used a random forest classifier (RFC) model to assess the precision of assigning samples to different enterotypes [40], thereby identifying the biomarkers that were important for enterotype classification.

### 2.9. Data Availability

The molecular sequence data were deposited in the NCBI Sequence Read Archive (SRA) database (accession number PRJNA76375 for dataset1; accession number PRJNA1091068 for dataset2).

## 3. Results

### 3.1. Identification and Diversity of Enterotypes in A. agrarius

The Bray–Curtis dissimilarity analysis at the genus level, accompanied by the Calinski–Harabasz (CH) evaluation and the corresponding silhouette scores, demonstrated the optimal partition of the gut microbiota communities of 94 individuals of *A. agrarius* into 2 clusters, as evidenced by the highest CH value observed (Figure 2A,B). Consequently, the entire sample cohort was stratified into two distinct enterotypes: enterotype 1 (Type1, N = 49) and enterotype 2 (Type2, N = 45). The principal coordinate analysis (PCoA) graphs underscored significant disparities in the structures of the intestinal community between these two clusters, as determined by ANOSIM (R = 0.596, *p* < 0.001) (Figure 2A). Moreover, intergroup Wilcoxon rank sum tests revealed substantial variations in the α diversity of the microbial communities between the identified enterotypes. Specifically, the Simpson index (*p* = 0.0016) and Shannon index (*p* < 0.001) of Type2 were significantly higher compared to those of Type1 (Figure 2C). This indicated that the Type2 enterotype harbored a more diverse microbial community in terms of both species richness and evenness compared to Type1.

Furthermore, we conducted a comparative analysis of the bacterial microbiota between the two enterotypes of *A. agrarius*. At the phylum level, Firmicutes and Bacteroidetes were found to be the most abundant phyla in both the Type1 and the Type2 enterotypes (Figure 3A). At the genus level, *Lactobacillus*, *Muribaculaceae*, and *Streptococcus* were identified as the predominant genera in both enterotypes (Figure 3B). Subsequently, we investigated the differences in these core gut microbiota between the different enterotypes. The results revealed that Type1 exhibited a higher abundance of Firmicutes, Actinobacteria, and Verrucomicrobia, while displaying a lower abundance of Bacteroidota, Campylobacterota, and Desulfobacterota at the phylum level (Figure 3C). At the genus level, Type1 exhibited a higher abundance of *Lactobacillus*, *Streptococcus*, *Enterorhabdus*, *Paraclostridium*, and *RF39*, along with a lower abundance of *Muribaculaceae*, *Helicobacter*, *Bacteroides*, *Alistipes*, *Colidextribacter*, *Roseburia*, *Prevotellaceae_UCG-003*, *Clostridia*, and *Acetatifactor* (Figure 3D). These findings delineate the distinct compositional profiles of the core gut microbiota between the Type1 and Type 2 enterotypes in *A. agrarius*.

A select set of characteristic bacterial genera was discerned via random forest analysis, serving as key discriminators that accounted for dissimilarities in the microbial composition among the distinct enterotypes (Figure 4A,B). Through 10-fold cross-validation, Lactobacillus, Streptococcus, and Muribaculaceae emerged as potentially pivotal markers for delineating the enterotype of *A. agrarius*.

### 3.2. Differences of Predicted Gene Functions between Enterotypes

The results of microbial gene function prediction showed that the functions of the microbial genes of *A. agrarius* were mainly concentrated in metabolism (Type1: 46.8%, Type2: 46.6%), genetic information processing (Type1: 19.1%, Type2: 18.9%), and environmental information processing (Type1: 15.4%, Type2: 14.9%) (Appendix A). At KEEG level 2, 11 predicted gene functions (i.e., membrane transport, carbohydrate metabolism, and nucleotide metabolism) were more abundant in Type1, while 14 predicted gene functions (i.e., energy metabolism, energy metabolism, and energy metabolism) were enriched in Type2 (Appendix A). At KEEG level 3, a total of 210 gene functions were significantly different between the 2 enterotypes. These results indicated that these two clusters had significantly different functions. Among these functional genes with a relative abundance of more than 1%, functions such as ABC transporters, DNA repair and recombination proteins, purine metabolism, function unknown, ribosome biogenesis, aminoacyl tRNA biosynthesis, glycolysis/gluconeogenesis, and pyruvate metabolism were significantly enriched in Type1, while those functions like the two-component system, bacterial motility proteins, methane metabolism, arginine and proline metabolism, and oxidative phosphorylation were enriched in Type2 (Figure 5). These results indicate that the gut microbiota of *A. agrarius* plays an important role in metabolic function and has different degrees of enrichment in different enterotypes.

### 3.3. Network Interactions of the Two Enterotypes

Network analysis revealed distinct patterns of microbial co-occurrence between the two enterotypes. Specifically, Type1 exhibited larger network diameters, higher graph density, greater average degree, and longer average path lengths. This trend suggested that the intestinal bacterial taxa within the Type1 network were more centrally located compared to Type2. Conversely, Type2 had a higher modularity and clustering coefficient, indicating the presence of more cohesive and functional microbial units within its network (Figure 6; Table 1). The differences in these topological properties reflected variations in the ecological structure and functional organization of the microbial communities between the two enterotypes.

### 3.4. Community Assembly Process in the Two Enterotypes of A. agrarius

To explore the assembly processes of the microbial communities in different enterotypes of *A. agrarius*, we first deployed the neutral model to assess the fit of the samples within different microbial communities. The results showed that the frequency of gut microbiota ASVs within Type1 fit the neutral model at 89.97% (R^2^ = 0.728, m = 0.002) and that of Type2 at 88.78% (R^2^ = 0.728, m = 0.002) (Appendix A; Figure 7A). This indicates that both deterministic and stochastic processes play a critical role in the formation of communities in the enterotype of *A. agrarius*.

The null model analysis showed that Type2 exhibited a higher betaMNTD value compared to Type1 (Figure 7B). This discrepancy indicates a greater degree of phylogenetic turnover among species within the microbial community of Type2 relative to Type1, suggesting a higher level of compositional heterogeneity in the microbial assemblages. Furthermore, the results showed that the bacterial community pooled across the two enterotypes was shaped primarily by homogeneous selection, homogenizing dispersal, and drift. However, the relative contribution of homogeneous selection was higher for Type1 (39.27%) than Type2 (38.84%), suggesting a slightly stronger influence of environmental factors driving bacterial community composition in Type1. Additionally, the relative contribution of homogenizing dispersal was higher for Type1 (32.27%) than Type2 (30.05%), indicating a higher level of microbial exchange or dispersal across individuals within Type1. Conversely, Type2 showed a higher relative contribution of drift (27.73%) compared to Type1 (25.18%), implying that stochastic processes play a relatively greater role in shaping the structure of the bacterial community in Type2 (Appendix A; Figure 7C). Overall, these findings underscore the nuanced interplay between deterministic and stochastic processes in governing the assembly and dynamics of bacterial communities within different enterotypes.

To examine the influence of external factors (altitude) and host intrinsic factors (body weight, body length, and body mass index) on the assembly of microbial communities among different taxa, we employed the Mantel test method to explore the relationship between these four distinct factors and the betaMNTD values. The analysis indicated that the betaMNTD values of both enterotypes did not show a significant correlation with body length (Type1: R^2^ < −0.001, *p* = 0.322; Type2: R^2^ < −0.001, *p* = 0.891) and body weight (Type1: R^2^ < −0.001, *p* = 0.256; Type2: R^2^ < −0.001, *p* = 0.249). However, it is noteworthy that altitude factors exhibited positive correlations with betaMNTD values in both enterotypes (Type1: R^2^ = 0.02, *p* < 0.001; R^2^ = 0.12, *p* < 0.001), as did body mass index (BMI) (Type1: R^2^ = 0.004, *p* = 0.017; R^2^ = 0.047, *p* < 0.001) (Figure 8A–D). These findings underscore the importance of environmental factors, particularly altitude and BMI, in influencing the structure and dynamics of microbial communities in the gut, potentially reflecting adaptations to specific environmental conditions and host physiological characteristics.

## 4. Discussion

Rodents play a crucial role in ecosystem functioning, with their gut microbiota significantly contributing to nutrient absorption, toxin degradation, and pathogen resistance [44]. Understanding the composition of gut microbiota in rodents is therefore of paramount importance for ecological management. In this study, we detected and analyzed the gut microbiota of 94 wild individuals of *A. agrarius* from 7 geographic populations, revealing, for the first time, the division of the gut microbiota of *A. agrarius* into two enterotypes. These two enterotypes exhibited distinct microbial diversity and predicted functions, interactions, and assembly processes, providing valuable information for the study of the types of gut microbial communities in wild animals.

Firstly, our results demonstrated that the gut microbiota composition of *A. agrarius* was dominated primarily by Firmicutes and Bacteroidetes, with notable abundances of *Lactobacillus* and *Muribaculaceae* genera; this is similar to the findings in other omnivorous mammals such as squirrels [45,46] and lemurs [24]. This suggests conservation of the composition of the gut microbiota throughout mammalian evolution, with selective colonization of those necessary microbes regardless of host identity [47,48]. Furthermore, the different compositional profiles of Type1 and Type2 enterotypes suggested differential ecological adaptations within the gut microbiota of *A. agrarius*. These adaptations could arise from various factors, including dietary preferences, habitat differences, and host genetics [49,50,51,52]. For instance, the higher abundance of *Lactobacillus* and *Streptococcus* in Type1 may reflect adaptations to specific dietary substrates or environmental conditions prevalent in the habitat of these individuals [53]. Conversely, the increased *Muribaculaceae* abundance and diversity observed in Type2 may be indicative of a broader dietary niche or more dynamic environmental interactions [54,55]. These biological differences serve as potential mediators of host specificity and warrant further investigation.

Beyond taxonomic differences, understanding the functional implications and network structures of enterotype variation is crucial for deciphering its significance in host health and ecology [56]. The prediction of microbial gene function revealed distinct functional spectra between the Type1 and Type2 enterotypes. While both types primarily focus on metabolism, genetic information processing, and environmental information processing, there are significant differences at both KEGG level 2 and level 3. The different metabolic capacities and network structures of the gut microbiota may influence their ability to perform specific metabolic functions, regulate host physiology, and respond to environmental changes. The Type1 enterotype exhibited enrichment in gene functions related to membrane transport, carbohydrate metabolism, and nucleotide metabolism, which is indicative of an emphasis on nutrient acquisition and processing. Conversely, the Type2 enterotype showed enrichment in gene functions associated with energy metabolism, bacterial motility proteins, and oxidative phosphorylation, suggesting a potentially more active metabolic state and higher energy production capacity. Furthermore, network analysis, in addition to serving as a valuable tool for studying host-associated microbial community patterns, was also used to detect key taxonomic groups that play essential ecological roles in the assembly of microbial communities or critical ecosystem functions [57]. Although many key taxonomic groups identified in our analysis were not assigned specific ecological roles within different enterotypes.

In numerous instances, deterministic and stochastic processes have been shown to interact synergistically rather than remain mutually exclusive during microbial community assembly [29,58]. Our study, employing neutral model analysis, revealed that the formation of gut microbiota communities in *A. agrarius*, regardless of enterotype, was influenced by a combination of stochastic processes, such as ecological drift and dispersal limitation, and deterministic factors, including environmental selection. However, there are subtle differences in the relative contributions of homogeneous selection, homogenizing dispersal, and drift between the two enterotypes, resulting in distinct assembly processes that govern the structure of the microbial communities in each. While both enterotypes were affected by homogeneous selection and dispersal, Type1 appeared to exhibit a marginally stronger influence on community composition, while Type2 leaned more towards stochastic processes such as ecological drift. This discrepancy can be attributed to variations in the habitat types inhabited by the sampled populations in our study. Moreover, the higher betaMNTD values observed in Type2, compared to Type1, indicated greater phylogenetic turnover among the species within the Type2 microbial communities, signifying a higher level of compositional heterogeneity. Consequently, Type2 harbored a more diverse and dynamically shifting microbial composition relative to Type1. This observation further elucidated why the microbial network in Type2 manifests higher clustering coefficients and modularity compared to Type1.

The βNTI metric was utilized to assess whether species interactions within a community aligned with their expected relationships based on their shared evolutionary history. In essence, it quantified the extent of the deviation between the relatedness of species in their shared evolutionary history and their actual relationships within the community. Initially, βMNTD was computed by calculating the branch lengths of bacterial phylogenetic trees, followed by the standardizing of the residuals between the observed values and expected values (obtained through 999 randomizations), which were then normalized to estimate βNTI. Subsequently, Mantel tests were employed to examine the correlation between βNTI and the pertinent factors. Our findings revealed a significant positive correlation between βNTI values and both altitude and body mass index (BMI) across the two intestinal types. This indicated that alterations in environmental factors induced substantial inconsistencies between the species interactions in the microbial communities and their expected relationships based on evolutionary history. Alternatively, it suggested a decay in similarity among gut microbial communities across individuals as a result of geographical distance. This finding aligned with conclusions drawn from studies on species such as house mice [59,60,61,62,63], pikas [20], woodrats [64], and wild leaf miners [31]. Variations in altitude and BMI may play a role in the systematic phylogenetic turnover differences or compositional disparities observed in the formation of the two enterotypes of gut microbiota. This could be attributed to the direct or indirect impact of altitude and BMI variations on host physiological conditions, metabolic activities, and immune system functionality [20,62,65,66]. These factors may affect the stability of the host’s gut environment and the availability of ecological niche space, resulting in alterations in the structure and function of microbial communities [67,68]. Additionally, changes in altitude and BMI may also influence the interaction between the host and the external environment, including diet, lifestyle, and microbial sources [7,69], thereby affecting the composition and diversity of microbial communities. Thus, variations in altitude and BMI may influence the assembly processes of microbial communities through multiple pathways, leading to differences in microbial community composition between the different enterotypes.

## 5. Conclusions

In summary, by employing an integrated approach drawing upon ecological and microbiological methodologies, we elucidated the presence of two distinct enterotypes within the gut microbiota of wild *A. agrarius* populations for the first time. Further investigations revealed distinct structural and functional attributes associated with these two enterotypes, with the assembly of microbial communities within each enterotype being influenced by both stochastic (body mass index, BMI) and deterministic (altitude) processes. Our findings underscore the significant relationship between gut microbiota and environmental gradients, thereby offering valuable insights into the implications for wildlife conservation and ecosystem management. Future research endeavors should involve longitudinal studies encompassing various seasons and altitudes, incorporating a larger sample size, and analyzing physiological parameters, such as host disease status, parasitic infections, and reproductive conditions, to comprehensively elucidate the precise associations and interactions between hosts and microbial communities.

## Figures and Tables

**Figure 1 microorganisms-12-00671-f001:**
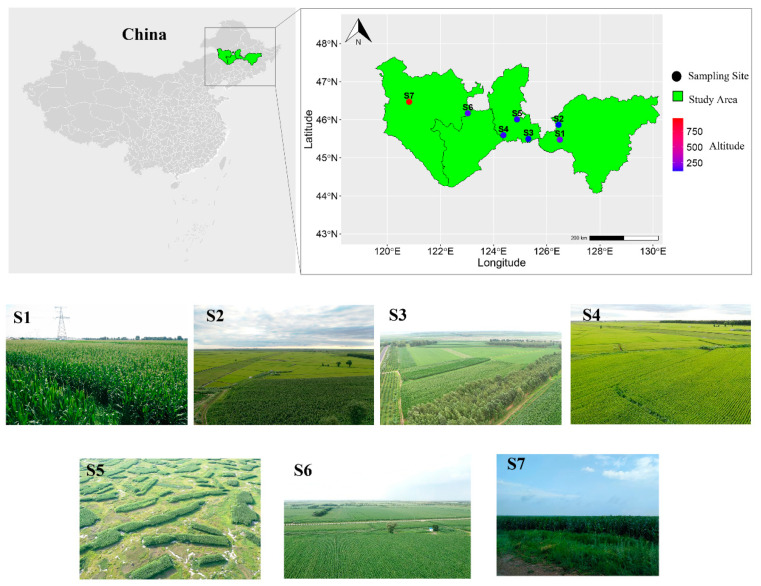
Sampling locations and habitat characteristics of wild *A. agrarius* across multiple geographic regions of China. The figure was generated using R software (version 4.3.2) based on a template map from the Chinese National Basic Geographic Information Center (http://ngcc.sbsm.gov.cn, accessed on 15 June 2020). Images of the habitats were taken by a DJI drone (model: Mavic Pro 2).

**Figure 2 microorganisms-12-00671-f002:**
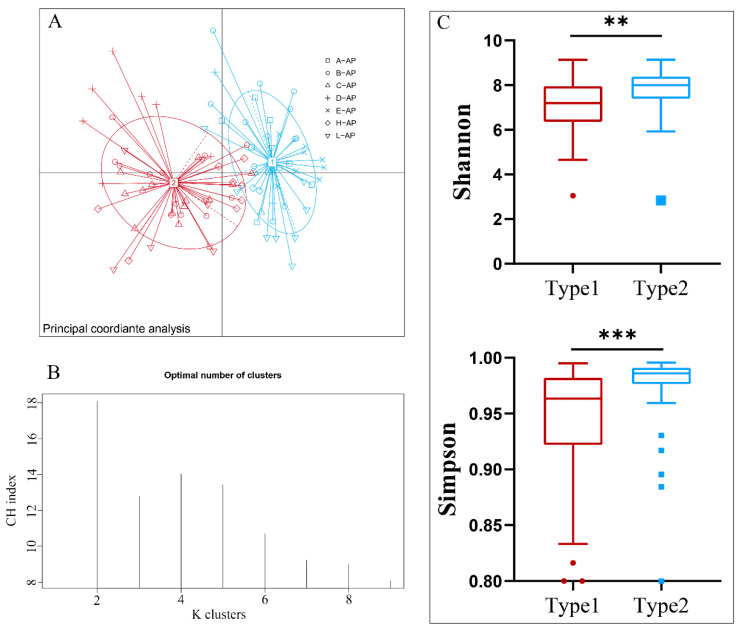
Identification and clustering analysis and alpha diversity of two enterotypes of *A. agrarius*. (**A**) Principal coordinate analysis (PCoA) plots based on Bray–Curtis dissimilarity metrics. (**B**) K-means partitions comparison and Calinski–Harabasz values calculation. The highest Calinski–Harabasz values indicate optimal clusters. (**C**) Mann–Whitney U test was used to test the differences of Shannon index and Simpson index between 2 types. Significant difference is indicated by ** *p* < 0.01, *** *p* < 0.001.

**Figure 3 microorganisms-12-00671-f003:**
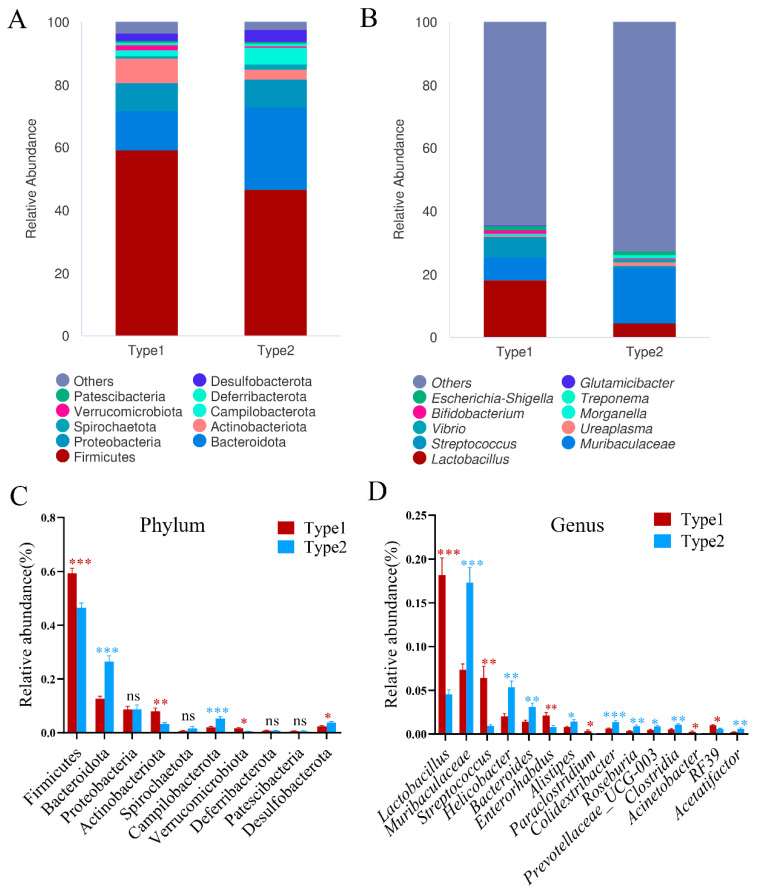
Taxonomic composition of 2 community types in *A. agrarius*. (**A**) Phyla level. (**B**) Genus level. (**C**,**D**) One-way with Tukey’s post hoc test was taken to test the compositional differences of 2 enterotypes at phyla and genus level. Significant difference is indicated by * *p* < 0.05, ** *p* < 0.01, *** *p* < 0.001, ns represents no significance.

**Figure 4 microorganisms-12-00671-f004:**
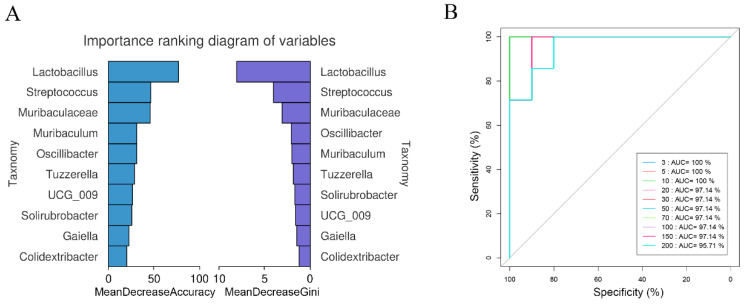
Representation of bacterial genera among sequence variants most informative in species-classification Random Forest Classifier (RFC) models. (**A**) The histogram is a ranking chart of the importance of variables. In the Accuracy chart, the horizontal coordinate represents the average declining accuracy, and the vertical coordinate is the top 10 important species. The Gin chart shows the average decline in the Gini index on the horizontal axis and the top 10 important species on the vertical axis. (**B**) The ROC curve serves as a measure to assess the credibility of random forest models established under varying feature counts (3, 5, 10, 30, 50, 70, 100, 150, and 200), with the false positive (Specificity) rate on the horizontal axis and the true positive (Sensitivity) rate on the vertical axis. Larger areas under the curve (AUC) indicate better model discrimination ability.

**Figure 5 microorganisms-12-00671-f005:**
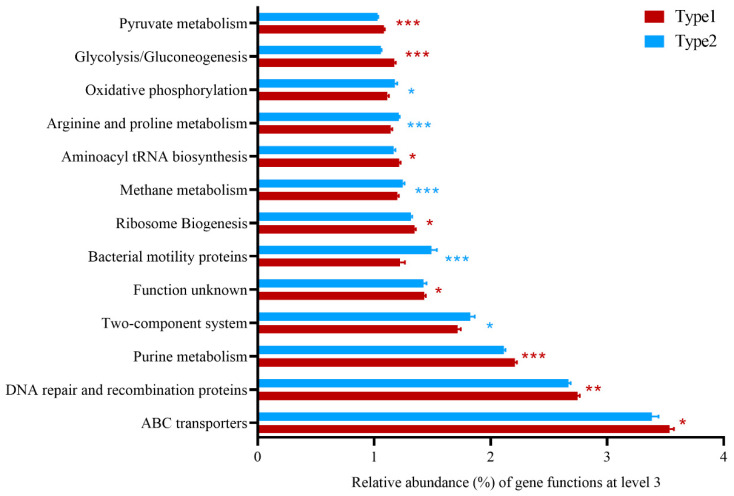
Functional prediction analysis of two community types in *A. agrarius*. Significant differences of predicted gene functions (Wilcoxon rank sum test) between clusters were evaluated at level 3. Significant difference is indicated by * *p* < 0.05, ** *p* < 0.01, *** *p* < 0.001.

**Figure 6 microorganisms-12-00671-f006:**
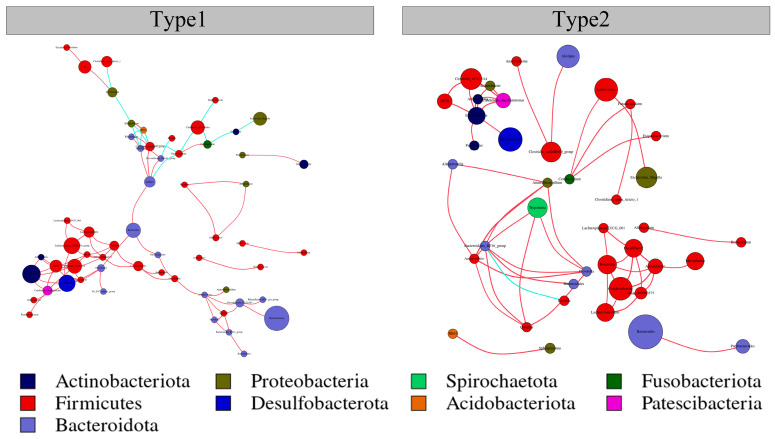
Metacommunity networks of 2 community types in *A. agrarius* based on Spearman’s correlation analysis. A connection between nodes indicates a strong (Spearman’s r > 0.6) and significant (FDR-corrected *p* value < 0.05) correlation. Co-occurrence networks for Type1 and Type2 are shown. The size of each node is proportional to the degree of the ASVs. The red links of 2 nodes represent positive correlation, and the blue links of 2 nodes represent negative correlation.

**Figure 7 microorganisms-12-00671-f007:**
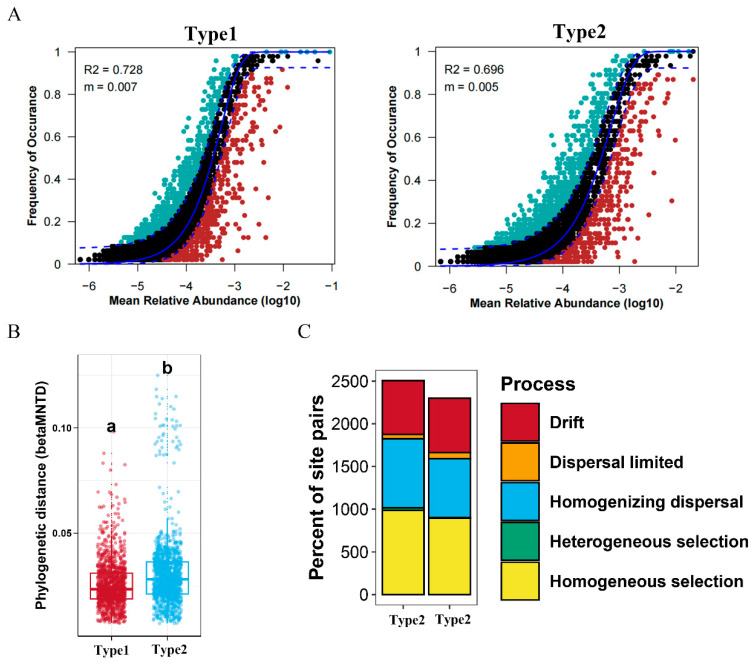
Assembly and ecological processes of gut microbiota for 2 community types in *A. agrarius*. (**A**) Fit of the neutral community model (NCM). Solid blue lines indicate the best fit to the model, while dashed blue lines represent 95% confidence intervals around its prediction. ASVs that occur more or less frequently than predicted are shown in different colors. The Nm value indicates the metacommunity size times immigration, and the R2 value indicates the fit to NCM (color figure online). (**B**) The betaMNTD values were calculated in different community types, the letter ‘a’ and ‘b’ indicate a significant difference between the two groups. (**C**) Summary of the relative contributions of the ecological processes that determine community assembly between the 2 community types.

**Figure 8 microorganisms-12-00671-f008:**
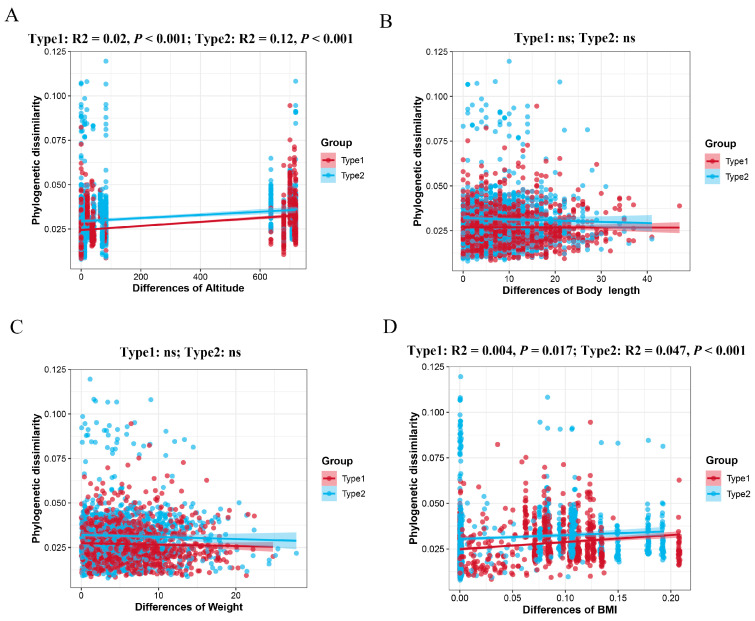
Effects of environmental factors on 2 enterotype assemblies of *A. agrarius*. Mantel analysis used to evaluate the correlation between the β-Nearest Taxon Index (βNTI) and the multiple environmental variables: (**A**) altitude, (**B**) body length, (**C**) body weight, (**D**) body mass index (BMI).

**Table 1 microorganisms-12-00671-t001:** Topological properties of co-occurrence networks for the 2 enterotypes.

	Network Diameter	Modularity	Clustering Coefficient	Graph Density	Average Degree	Average Path Length
Type1	12	0.706	0.392	0.018	3.469	5.109
Type2	4	0.768	0.489	0.010	2.061	1.673

## Data Availability

Molecular sequence data were deposited in the NCBI Sequence Read Archive (SRA) database (accession number PRJNA1019510 for dataset1; Dataset 2 is being uploaded for public release).

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
