# Peer review of "Distinct Gut Microbial Enterotypes and Functional Dynamics in Wild Striped Field Mice (Apodemus agrarius) across Diverse Populations"

_microorganisms, 2024, doi:10.3390/microorganisms12040671_

Round 1
Reviewer 1 Report
Comments and Suggestions for Authors
The paper investigates the gut microbiota of wild populations of Apodemus agrarius, categorizing them into two distinct enterotypes characterized by Lactobacillus and Muribaculaceae. It highlights the differential ecological adaptations, functional profiles, and assembly processes between these enterotypes, influenced by both stochastic (e.g., BMI) and deterministic (e.g., altitude) factors. The study underscores the importance of understanding the evolutionary dynamics of rodent gut microbiota and suggests future research directions to comprehensively elucidate host-microbiota interactions in wild populations.
The following suggestions are proposed to the authors to improve the presentation of the paper.
Title: Consider, a revision of the title to "Distinct Gut Microbial Enterotypes and Functional Dynamics in Wild Striped Field Mice (Apodemus agrarius) Across Diverse Populations", in order to to enhance clarity and specificity.
Abstract: Please specify the geographic regions sampled.
Keywords: The keyword “rodents” should be replaced by Apodemus agrarius or just Apodemus agrarius to be added in the keywords. Erase the word “introduction” from the keywords.
Introduction:
- Provide a brief definition of the term “Community assembly” that you use as keyword.
- "enterotype" is sometimes written with a hyphen ("-") and sometimes without. Choose one format and apply it consistently throughout the text.
- Provide more detailed explanations or definitions for terms and concepts introduced, such as "niche differentiation" and "neutral processes."
2.1. Sample Collection:
- The elevation values for sampling sites S4 and S5 are identical, suggesting a potential error. Please clarify.
4. Discussion:
-Provide a more detailed explanation of the Mantel test analysis methodology, including the interpretation of betaMNTD values and their significance in assessing microbial community similarity across different environmental and host-related gradients.
5. Conclusions:
- Conclude with a brief summary of the broader implications of the study findings for understanding the evolutionary ecology of rodent gut microbiota and their potential relevance for wildlife conservation and ecosystem management.
- Highlight the novelty and significance of the study.
Reviewer 2 Report
Comments and Suggestions for Authors
This interesting study examined the gut microbiota of 94 wild striped field mice (Apodemus agrarius) from seven geographic populations identifying two distinct enterotypes (Type1 and Type2) with unique microbial compositions and functions. They found differences in metabolic processes between the enterotypes, as well as a mix of stochastic and deterministic factors shaping microbiota assembly. Correlation analysis linked elevation and BMI to microbial community turnover, suggesting environmental and physiological factors influence gut microbiota diversity in these mice. The manuscript is very well-structured and written and provides very valuable information. Please find my brief comments below.
Specific comments:
Abstract:
· Line 22: Introduction section must be separated.
· Line 24: “enterotype”, in singular.
· Lines 48-50: Please unify. Genus and species in brackets in all or none.
· Lines 74-75: References to the 17 zoonotic pathogens are recommended.
· Line 79: “spatiotemporal” I understand that there were spatial differences in sample collection, but temporal differences are not clear (even dataset 1 is from July to August 2020 and dataset the same period but in 2021). This is a cross-sectional study, and animals were not followed to analyze microbiome variability, so this term may be confusing and misleading.
Materials and Methods
· Line 93: In any part of the sample collection section, the authors explain what sample they are collecting. Even gut microbiota is mentioned, there are several ways to study gut microbial communities, including rectal swabs collection, natural voided faeces, or even whole gut examination after necropsy. Authors must therefore explain the origin of the sample in the manuscript. On the other hand, the country (China) must also be mentioned as they mention different locations, but the reader must look at Figure 1 to get this information.
· Line 107: Wu et al. (2024). This paper cannot be consulted.
· Line 112: A. agrarius must be in italics.
· Line 121: SYBR instead of SYB?
· Line 145: sequence variants of amphibians?? ASVs: amplicon sequence variant.
· Line 209: “Wilcoxon rank sum test” instead of “Wilcox rank sum test”
Results
· Figure 2: Shannon and Simpson should be in capital letters.
· Figure 3: I suggest enlarged figures C and D.
· Figure 6: in pikas??
Discussion
· Line 368-370: Again, A. agrarius must be in italics.
· Line 391: “type 1 and Type 2 enterotypes” unify.
Conclusions
· Line 452: in rodent gut microbiota of rodents?
